# A protease protection assay for the detection of internalized alpha-synuclein pre-formed fibrils

Timothy S. Jarvela, Kriti Chaplot, Iris Lindberg*

Department of Anatomy and Neurobiology, University of Maryland-Baltimore, Baltimore, MD, United States of America

* ilindberg@som.umaryland.edu

## Abstract

Alpha-synuclein pre-formed fibrils (PFFs) represent a promising model system for the study of cellular processes underlying cell-to-cell transmission of alpha-synuclein proteopathic aggregates. However, the ability to differentiate the fate of internalized PFFs from those which remain in the extracellular environment remains limited due to the propensity for PFFs to adhere to the cell surface. Removal of PFFs requires repeated washing and/or specific quenching of extracellular fluorescent PFF signals. In this paper we present a new method for analyzing the fate of internalized alpha-synuclein. We inserted a tobacco etch virus (TEV) protease cleavage site between alpha-synuclein and green fluorescent protein and subjected cells to brief treatment with TEV protease after incubation with tagged PFFs. As the TEV protease is highly specific, non-toxic, and active under physiological conditions, protection from TEV cleavage can be used to distinguish internalized PFFs from those which remain attached to the cell surface. Using this experimental paradigm, downstream intracellular events can be analyzed via live or fixed cell microscopy as well as by Western blotting. We suggest that this method will be useful for understanding the fate of PFFs after endocytosis under various experimental manipulations.

## Introduction

Cell-to-cell transmission of proteopathic aggregates is increasingly understood to represent the underlying cause of the stereotypical spread of α-synuclein pathology observed in Parkinson's disease [1]. A variety of model systems have been developed to replicate this process *in vivo* [2–7] as well as in primary and immortalized cell culture [8–10]. Taken together, these experimental models have identified specific cellular components responsible for the internalization and transmission of α-synuclein.

Pre-formed fibrils (PFFs) represent a molecular form of α-synuclein thought to represent a good proxy for the cytotoxic species produced *in vivo* (reviewed in [10]). However, it has been difficult to develop experimental methods that permit the biochemical discrimination of intracellular PFFs from those remaining extracellularly, for example PFFs coating the cell surface.

**Data Availability Statement:** All relevant data are within the manuscript and figures. Raw gel and blot data are in Supporting information.

**Funding:** This work was supported by NIH grant AG062222 to IL. The funders had no role in study

   

design, data collection and analysis, decision to publish, or preparation of the manuscript. There was no additional external funding received for this study.

**Competing interests:** The authors have declared that no competing interests exist.

This problem is especially difficult to overcome given the known stickiness of synuclein aggregates. While trypan blue treatment of GFP-tagged synuclein has been used to quench the GFP signal from surface-attached PFFs [10], this method cannot be used in concert with immuno-cytochemical or biochemical analyses, or with prolonged chase periods following the quench. Because of the limitations of the current cell biology toolkit, our knowledge of the key processes and molecular players involved in the fate of pathological extracellular α-synuclein seeds following binding and internalization remains limited.

We here present a method for the specific detection of internalized α-synuclein PFFs. The assay provides an intact *vs* cleaved protein readout in response to incubation with or without TEV protease. With this method, intracellular α-synuclein PFFs are protected against cleavage due to the impermeability of the plasma membrane. The ability of membranes to block protease cleavage of target proteins has frequently been used to determine the subcellular localization and topology of transmembrane domain-containing proteins ([11]; reviewed in [12]). Here, we employ GFP-tagged α-synuclein PFFs containing an engineered TEV-P cleavage site to efficiently discriminate between intracellular and extracellular PFFs.

## Methods

### Plasmids

pET21a-α-synuclein, encoding human α-synuclein, was a gift of the Michael J. Fox Foundation (Addgene plasmid #51486). The pRK172-mouse α-synuclein-GFP plasmid was a kind gift of Dr. Virginia Lee (University of Pennsylvania) [10]. The tobacco etch virus protease (TEV) site (ENLYFQG) was inserted following the 6x His tag, in frame with the start of the GFP sequence. A forward primer (5'-TAT TTT CAG GGC ATG GTG AGC AAG GGC GAG-3') and reverse primer (5'- AAG ATT CTC GAG ATG GTG ATG GTG ATG G-3') were combined using the Q5 site-directed mutagenesis protocol (New England Biolabs, Ipswitch, MA). The resulting plasmid, pRK172-α-syn-TEV-GFP, was sequenced and transformed into competent BL21(DE3) cells (Thermo Fisher Scientific, Waltham, MA, Cat# EC0114) for protein production.

### Protein expression and purification

Protein production was performed using a protocol adapted from [13]. A 5 ml starter culture of transformed bacteria was incubated in LB broth (Sigma, St. Louis, MO) with ampicillin (50 μg/ml) at 37˚C overnight. One liter of autoinduction media (10 g tryptone, 5 g yeast extract, 2 ml 1 M $MgSO_4$, $H_2O$ to 930 ml) was autoclaved and completed by adding 50 ml of filter-sterilized 20x NPS (20x NPS: 0.5 M $(NH_4)_2SO_4$; 1 M $KH_2PO_4$, 1 M $Na_2HPO_4$) and 20 ml of filter-sterilized 50x 5052 (50x 5052: 50 g glycerol, 5 g glucose, 20 g lactose; $H_2O$ to 200 ml) prior to inoculating autoinduction media with starter culture. The culture was then grown in a shaking incubator at 30˚C for 24 h. Cells were harvested by centrifugation at 4000 x *g* at 4˚C for 20 min and the pellet frozen at -20˚C for later use.

Pellets were thawed at room temperature and resuspended in a 10% volume of osmotic shock buffer (100 ml per 1000 ml culture; 30 mM Tris pH 7.2, 2 mM EDTA, 40% sucrose). The suspension was centrifuged at 9000 x g at 20˚C for 20 min. The supernatant was discarded (a small amount of α-synuclein-GFP remains in the supernatant at this point and can be recovered by dialysis into buffer A) and the pellet was resuspended in 40 ml of ice-cold $H_2O$. A saturated $MgCl_2$ solution was added to the resuspension at a 1:1000 dilution (40 μl). The mixed suspension was incubated on ice for 3 min, and then centrifuged at 4˚C for 30 min at 9000 x g. The visibly green supernatant was collected; at this point, it was highly fluorescent under UV light. The supernatant was loaded onto a 1 ml HiTRAP DEAE column (GE Healthcare,

Boston, MA) for ion exchange chromatography using an GE Akta Purifier (buffer A: 20 mM Tris-HCl, pH 7.4 (at 4°C), buffer B: 20 mM Tris, 1 M NaCl, pH 7.4 (at 4°C)). The purification protocol was as follows: 1 column volume (CV) buffer A; 2 CV 5% buffer B; 2 CV ramp to 10% buffer B; 2 CV 10% buffer B; 18 CV ramp to 80% buffer B, 2.5 ml fractions; 2 CV ramp to 95% buffer B; and 3 CV at 95% buffer B to finish washing. Eluted fractions can be identified using a hand-held UV-A light (wavelength 315–400 nm); fractions containing GFP appear yellow under normal light and fluoresce green under ultraviolet light. Alternatively, the elution peak can be determined by measuring absorbance at 488 nm. An aliquot of each fraction was analyzed by SDS polyacrylamide electrophoresis (SDS-PAGE; see below) followed by Coomassie staining, and fractions with strong α-syn-TEV-GFP bands were combined and further purified by His-tag affinity chromatography. Human untagged α-synuclein used for *in vitro* fibrillation assays was prepared in a similar manner.

### PFF generation

Lyophilized α-syn-TEV-GFP protein was resuspended in a small volume of Dulbecco's PBS (dPBS), filtered through a 0.45 μm filter, and diluted to 7.5 mg in a final volume of 1 ml in dPBS diluted 1/3 with water, as described in [10]. The suspension was then shaken for 7 days at 37°C in a 1.5 ml microcentrifuge tube [10]. After 2 days the mixture became turbid. On day 7, the mixture was vortexed vigorously to create a homogeneous suspension that was then aliquoted into 50 μl single-use aliquots and frozen at -80. For experiments, one aliquot was rapidly thawed and kept at room temperature. Ten μl of fibril suspension were added to 490 μl dPBS and sonicated using a Branson sonicator with a 0.125 inch microtip, 10% power, 50% duty cycle, for 90 sec at room temperature. PFFs were not placed on ice, as that has been shown to lead to dissociation of fibrils [14]. For the experiment shown in **Fig 6B**, α-syn-GFP protein lacking the TEV site [10] was prepared as described above, fibrillated, and sonicated similarly.

### Cell culture

Neuro2A cells (ATCC #CCL-131; Gaithersburg, MD) were grown at 37°C at 5% $CO_2$ in Opti-MEM/F12 (50/50; ThermoFisher Scientific) with 5% fetal bovine serum (Atlanta Biologicals, Flowery Branch, GA).

### Protease protection assay

TEV protease (TEV-P) was purchased from NEB (Cat#P8112S). PFFs were prepared as above and premixed 1:10 (for a final concentration of 15 μg/ml) into OptiMEM (ThermoFisher Scientific). The PFF/OptiMEM mixture was then added to cells in various volumes of OptiMEM (10 μl for microwell inserts, 100 μl for 48-well plates) and incubated for approximately 1h. Cells were gently rinsed once with OptiMEM, and fresh OptiMEM was added. TEV-P (10,000 units/ml stock) was diluted 1:10 in OptiMEM; this TEV-P/OptiMEM mixture was then diluted a further ten-fold into each well to a final concentration of 100 units/ml (1 μl for microwell inserts, 10 μl for 48-well plates). Controls were treated similarly, but using enzyme storage buffer (50 mM Tris-HCl, 250 mM NaCl, 1 mM EDTA, 50% glycerol) in place of TEV-P. The cells/PFFs were then incubated in the presence of TEV-P or buffer for either 30 min (imaging experiments) or 60 min (Western blot experiments).

### Lysotracker experiment

Cells were grown and treated with PFFs as in the protease protection assay with the following addition: during the TEV-P incubation, LysoTracker® Red DND-99 (Molecular Probes, Life

Technologies) was added at 50 nM final concentration. At the end of the TEV-P and Lyso-tracker incubation, cells were rinsed once in prewarmed media without TEV-P or Lysotracker, fresh media were added to the cells, and images were acquired as described below.

## Primary neuron experiment

Primary hippocampal neurons were prepared as described [15]. Twenty thousand cells were plated on poly-L-lysine coated coverslips placed in a 24-well plate and allowed to differentiate in NeuroBasal medium containing 1x Glutamax, 20 µg/ml gentamycin and 1x B27 supplement for 7 days (all reagents were obtained from Sigma). The cells were then treated with 5 µg/ml GFP-TEV-α-syn PFFs for 1 h, followed by treatment with TEV-P for 30 min, and rinsed with culture medium after each step. After 4 more days of incubation in culture, cells were fixed in 4% paraformaldehyde for 15 min. Samples were stained with chicken anti-GFP antibody (AVES, Cat#GFP-1010, 1:1000) overnight at 4˚C, followed by anti-chicken IgY linked to Alexa Fluor 647 (Invitrogen, Cat#A32933, 1:1000) and Hoescht stain for 1 h at room temperature. Samples were imaged using a Leica SP8 40x oil immersion objective.

## Confocal microscopy

Neuro2A cells were grown in 4-well micro-inserts (Ibidi, Planegg, Germany; #80409) attached to µ-Dish 35-mm gridded polymer cover-slipped dishes (Ibidi, #81166) to allow concurrent imaging of multiple conditions as well as to minimize the volume of reagents required. Cells were seeded at approximately $2x10^4$ cells per well, in 10 µl of media, one day prior to PFF treatment. Excess media was added around the micro-insert as an evaporation barrier, as per the manufacturer's protocol. PFFs and TEV-P were added as described above. Post-TEV-P treatment images of GFP fluorescence were acquired (excitation: 488 nm; emission: 500–550 nm) and 10 µl of 2 mM trypan blue in dPBS was then added to each microwell to quench extracellular fluorescence. After 1 min a post-quench image was acquired both for GFP fluorescence and for trypan blue fluorescence (excitation: 561 nm; emission: 570–640 nm). For Lysotracker experiments, Lysotracker red images were acquired with a 561 nm excitation laser and a 570–640 nm emission filter. Live cell confocal microscopy was performed on a Nikon W1 spinning disk confocal system (Nikon Ti2 inverted microscope; Hamamatsu sCMOS camera), at the Confocal Microscopy Core Facility, University of Maryland-Baltimore. Acquisition was controlled using Nikon Elements software. Cells were incubated at 37˚C and under 5% $CO_2$ in Opti-MEM during imaging. Image analysis was performed with ImageJ using the "Analyze Particles" plugin to measure total fluorescence signal for a 3D image stack. For Lysotracker experiments, the 3D object counter plugin was run on background-subtracted images to determine discrete GFP-positive objects for each image. These objects were then scored for the presence or absence of Lysotracker red signal. For comparison to PFFs lacking a TEV cleavage site, the total GFP signal of 3D confocal slices were summed and normalized to the total number of cells per image by counting nuclei.

## Viability analysis

Neuro2A cells were plated at 4000 cells/ well in complete medium in a 96-well plate. Twenty-four h later, the culture medium was removed and replaced with 50 µl of OptiMEM. Fifty µl of OptiMEM containing either TEV-P, Triton X-100, or TEV-P buffer (final concentrations 100 U/ml, 1%, and 2% respectively) were then added to the appropriate wells in quadruplicate, and the plate incubated at 37˚C for 30 min. Buffer-treated and untreated cells were used as controls. The cells were rinsed once with OptiMEM, the medium replaced with complete Neuro2A medium, and the cells allowed to recover in the incubator for 24 h. The culture medium

was then removed and replaced with 50 μl of fresh Neuro2A medium. Fifty μl of Neuro2A medium containing 10 μl of WST-1 reagent (Sigma) were then added to each well and the plate incubated at 37˚C for 2 h. The absorbance of 100 μl of the medium was then measured in a 96-well plate using a BenchmarkPlus microplate spectrophotometer (Biorad). The absorbance at 690 nm, and the absorbance of the medium-only blank, were subtracted from the absorbance at 450 nm. The percent survival was computed by normalizing the absorbance values for each well with the average values from untreated wells. One-way ANOVA and Tukey's multiple comparisons tests were performed to determine statistical significance.

## SDS-PAGE and western blotting

Neuro2A cells were seeded at approximately $1x10^5$ cells per well in a 48-well tissue culture plate (Corning). After 24 h, media were replaced with prepared PFF media (as above, 100 μl per well) and returned to incubate for 1 h. TEV-P treatment was performed as described above, and cells were incubated for 1 h. Cells were then quickly denatured by adding 100 μl of 2x sample buffer (100 mM Tris-HCl, 8% SDS, 4% β-mercaptoethanol, 24% glycerol, 0.02% bromophenol blue, pH 6.8) to the wells. Samples were then heated at 95˚C for 10 min prior to SDS-PAGE electrophoresis. Electrophoresis using Mini-PROTEAN AnykD precast gels (BioRad, Richmond, CA) was performed using 15 μl of the lysis solution. Proteins were transferred to nitrocellulose using a Trans-Blot Turbo Transfer System (BioRad), using the mixed MW protocol (7 min, 1.3 amps). Membranes were blocked in 5% blotting grade non-fat dry milk blocker (BioRad) in Tris-buffered saline (TBS) containing 0.1% Tween 20 for 30 min and then incubated with mouse anti-α-synuclein antibody (BD Biosciences, San Jose, CA, cat. #610787) (1:2000 dilution from stock, in blocking buffer) and rocked overnight at 4˚C. Blots were washed 3x for 5 min each in TBS without Tween and incubated with goat anti-mouse IgG coupled to horseradish peroxidase (BioRad, #170656) at a 1:3000 dilution in blocking buffer for 1 h. Blots were washed again 3x in TBS and incubated in Clarity ECL (BioRad) prior to imaging using a BioRad Gel Doc Imager and quantification using Image Lab 6.0 software (BioRad).

## Electron microscopy

EM preparation was performed as described in [16]. Copper grids were floated on a droplet of water for 1 min and excess water was wicked away; this wash was repeated once. The grid was then floated on a suspension containing α-synuclein monomers, fibrils, or sonicated PFFs (each at 150 μg/ml in dPBS) for 1 min. Excess liquid was wicked away and the grid was floated twice in 2% uranyl acetate for one min each. Excess uranyl acetate was wicked away and grids were allowed to dry. Grids were examined in a Tecnai T12 transmission electron microscope (ThermoFisher Scientific; formerly FEI Co.; Hillsboro, OR) operated at 80 kEV. Digital images were acquired using a bottom-mounted CCD camera (Advanced Microscopy Techniques, Corp, Woburn, MA) and AMT600 software. Quantification of fibril length was performed using ImageJ.

## Thioflavin T assay (ThT)

Seeding was performed by adding 5 μl of 150 ng/μl sonicated PFFs or monomers in Dulbecco's PBS (dPBS) to each individual fibrillation reaction. Fibrillations were performed as previously described [17]. Briefly, in a round bottom 96-well polypropylene dish (Falcon), untagged human α-synuclein (100 μg/well)—purified as previously described [17]—and 10 μM ThT in dPBS (Sigma-Aldrich, St. Louis, MO) were combined in a final volume of 95 μl of dPBS. Seeds (5 μl PFF or monomer, in dPBS at 150 μg/ml) or an equivalent volume of PBS (non-seeded

control) were added to each well, and then a single 3/32inch polytetrafluoroethylene bead (McMaster-Carr, Atlanta, GA) was added to each well and the plate sealed with foil. The plate was then incubated at 37˚C while pulse shaking at 1000 RPM on a Microplate Genie Pulse Digital Mixer (Scientific Industries, Bohemia, NY). The plate fluorescence was measured at the indicated time points using a SpectraMAX M2 spectrophotometer (Molecular Devices, San Jose, CA), with an excitation peak at 450 nm and emission peak at 485 nm and a 475 nm band pass cutoff, with reads from the bottom. For each time point, three independent reads were taken at 20 sec intervals and values were averaged together. Each condition had 4 technical replicates, and the experiment was repeated once.

## Statistical analysis

GraphPad Prism 8 (GraphPad Software, La Jolla, USA) was used for statistical analysis and figure preparation; statistical details are given in figure legends.

## Results

To generate the construct used in this assay, a seven-residue TEV protease consensus site linker (ENLYFQG) was inserted after the 6x His-tag C-terminally attached to α- synuclein; and before the GFP tag (α-syn-TEV-GFP) (**Fig 1A**). The general strategy of the assay involves incubation of cells with α-syn-TEV-GFP fibrils, which are freshly sonicated to form preformed fibrils (PFFs), to permit internalization. After a short incubation, TEV protease (TEV-P) is added and cells are further incubated to permit cleavage of GFP from α-synuclein (**Fig 1B**). As proof-of-concept, purified α-syn-TEV-GFP PFFs were incubated with TEV-P *in vitro*. The GFP in these PFFs can be readily cleaved from α-synuclein, as evidenced by a single band of high molecular weight (approximately 45 kDa) in untreated wells and smaller bands of GFP (25 kDa) and α–synuclein (17 kDa) visible in TEV P-treated lanes (**Fig 1C**).

While Karpowitz *et al*. have shown that mouse α-synuclein with a C-terminal GFP-tag can form fibrils, and that GFP-tagged PFFs are able to spread pathology [10], we verified that the addition of a TEV cleavage site did not reduce the ability of the fusion construct to form fibrils. Using electron microscopy, we first confirmed the morphology of the α-syn-TEV-GFP monomers, fibrils, sonicated PFFs, and TEV-incubated PFFs (**Fig 2A**). These data support the efficacy of the sonication procedure and show that TEV treatment does not grossly alter PFF morphology. These data are quantified in **Fig 2B**, in which PFF lengths of TEV-treated and non-TEV-treated PFFs are compared. Others have noted that using PFFs with an average size of <50 nm is important to improve seeding both *in vivo* and *in vitro* [18]. The new α-syn-TEV-GFP PFFs were also capable of seeding fibril formation. The insertion of the TEV protease consensus site does not interfere with the seeding of untagged human α-synuclein when assayed using a thioflavin T fibrillation assay. When 100 μg of α-synuclein monomers were seeded with 750 ng of α-syn-TEV-GFP PFFs, the lag phase was reduced compared to wells seeded with monomers or unseeded control wells (**Fig 2C**). These data confirm that α-syn-TEV-GFP PFFs made from the construct shown in **Fig 1A** are fully competent in initiating the fibrillation of monomeric α-synuclein, a prerequisite for serving as a model system for α-synuclein transmission experiments.

Following these successful proof-of-concept and quality control assays, we tested the ability of TEV-P to cleave PFFs attached to cells. Since surface-associated PFFs are exposed to medium TEV-P, while internalized PFFs are not, treatment of cells with TEV-P can be used to distinguish internalized α-synuclein forms. We confirmed internalization of α-syn-TEV-GFP using the trypan blue quench technique, which employs this dye to quench extracellular surface-associated GFP [10]. α-syn-TEV-GFP PFFs were added to cells grown in imaging-

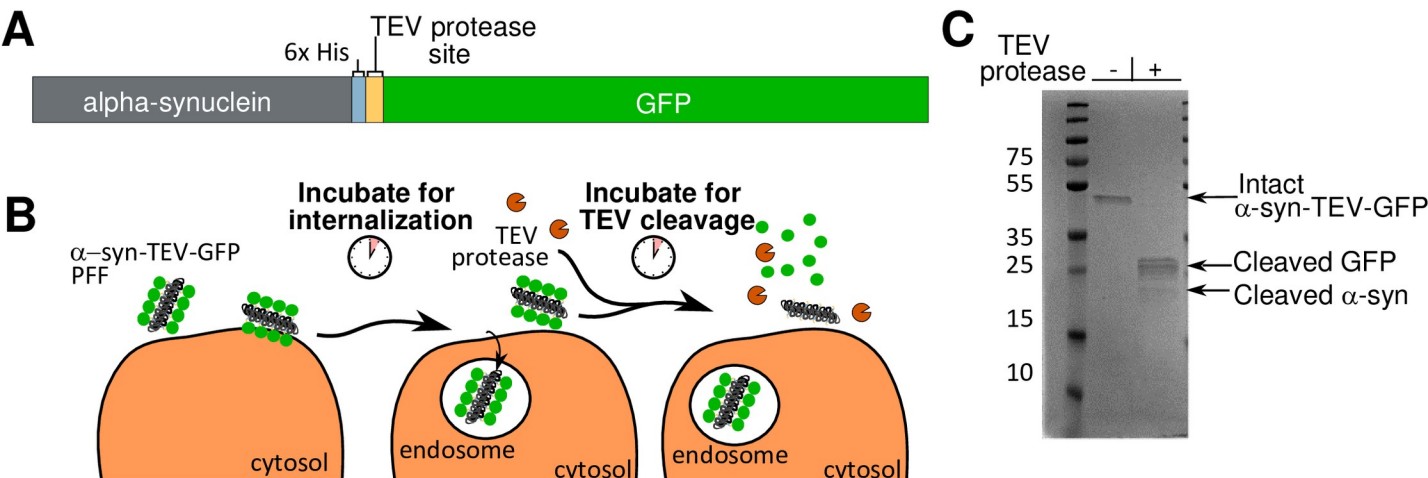

**Fig 1. Creation and schematic design of a TEV protease-sensitive α-synuclein-GFP fusion protein. (Panel A)** A schematic diagram of the fusion protein construct: mouse α-synuclein, 6x His, TEV protease site, and GFP. **(Panel B)** Schema of the protease protection assay. Pre-formed fibrils (black) containing GFP (*green*) are incubated with cells for 1 h to permit endocytosis into cells. TEV protease (*orange*) is added and incubated with cells for 30–60 min for optimal fusion protein cleavage. GFP is cleaved from PFFs exposed to TEV protease within the media and on cell surfaces. Internalized α-syn-TEV-GFP PFFs are protected from TEV cleavage. **(Panel C)** Coomassie-stained gel showing α-syn-TEV-GFP PFFs incubated *in vitro* with and without TEV protease. Sonicated PFFs were diluted to 75 μg/ml in 100 μl of dPBS in two separate tubes. Five μl of TEV protease stock or buffer were added, and tubes incubated for 1 h at 37°C. One hundred μl of 2x sample buffer were then added and samples heated prior to SDS-PAGE and Coomassie staining.

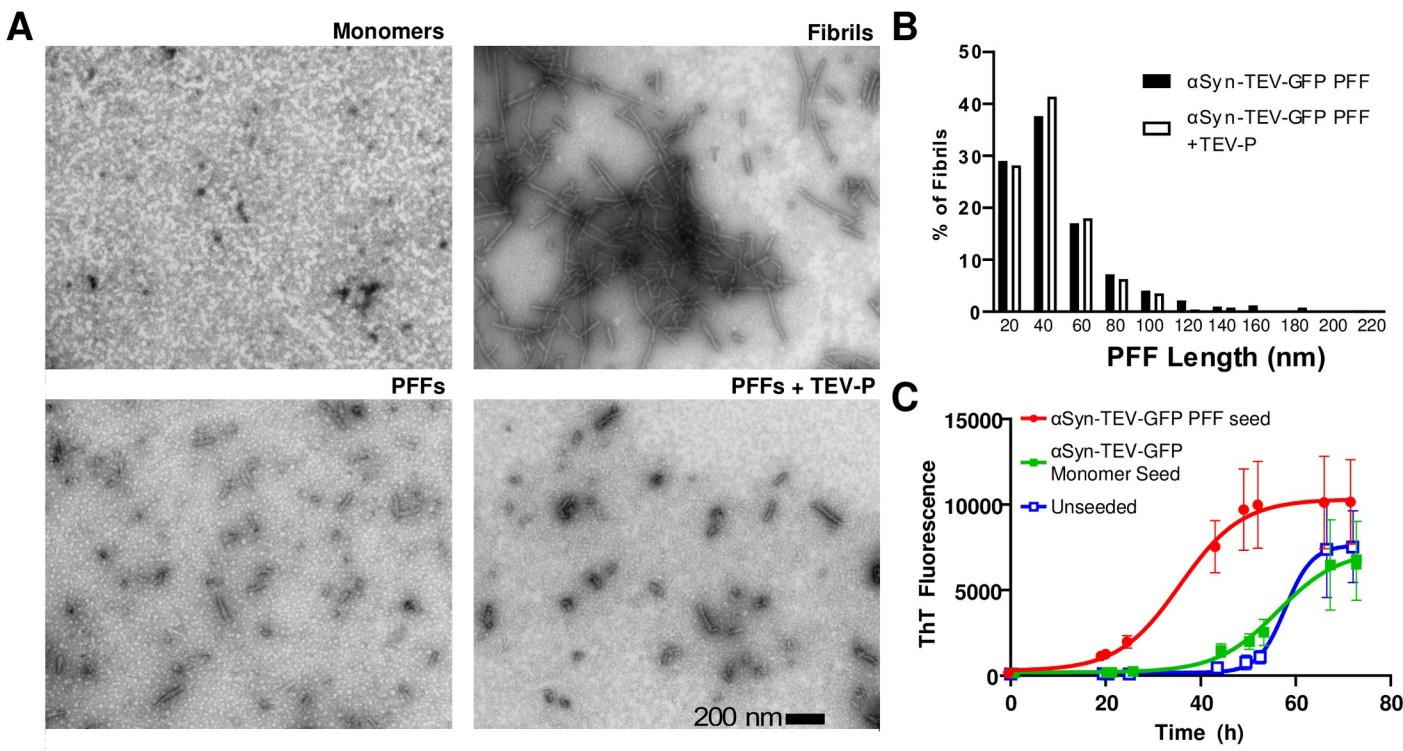

**Fig 2. Characterization of α-syn-TEV-GFP pre-formed fibrils and seeding efficacy. (Panel A)** Electron microscopy images of negatively-stained α-synuclein monomers, intact fibrils, sonicated fibrils, and sonicated fibrils following TEV treatment. (Scale bar = 200 nm). **(Panel B)** Histogram of fibril size distribution after sonication with and without TEV treatment (bin size 20 nm; >200 fibrils counted for each condition). **(Panel C)** α-syn-TEV-GFP PFFs are competent to seed the fibrillation of α-synuclein monomers. Thioflavin T fluorescence, generated following fibril formation, is shown.

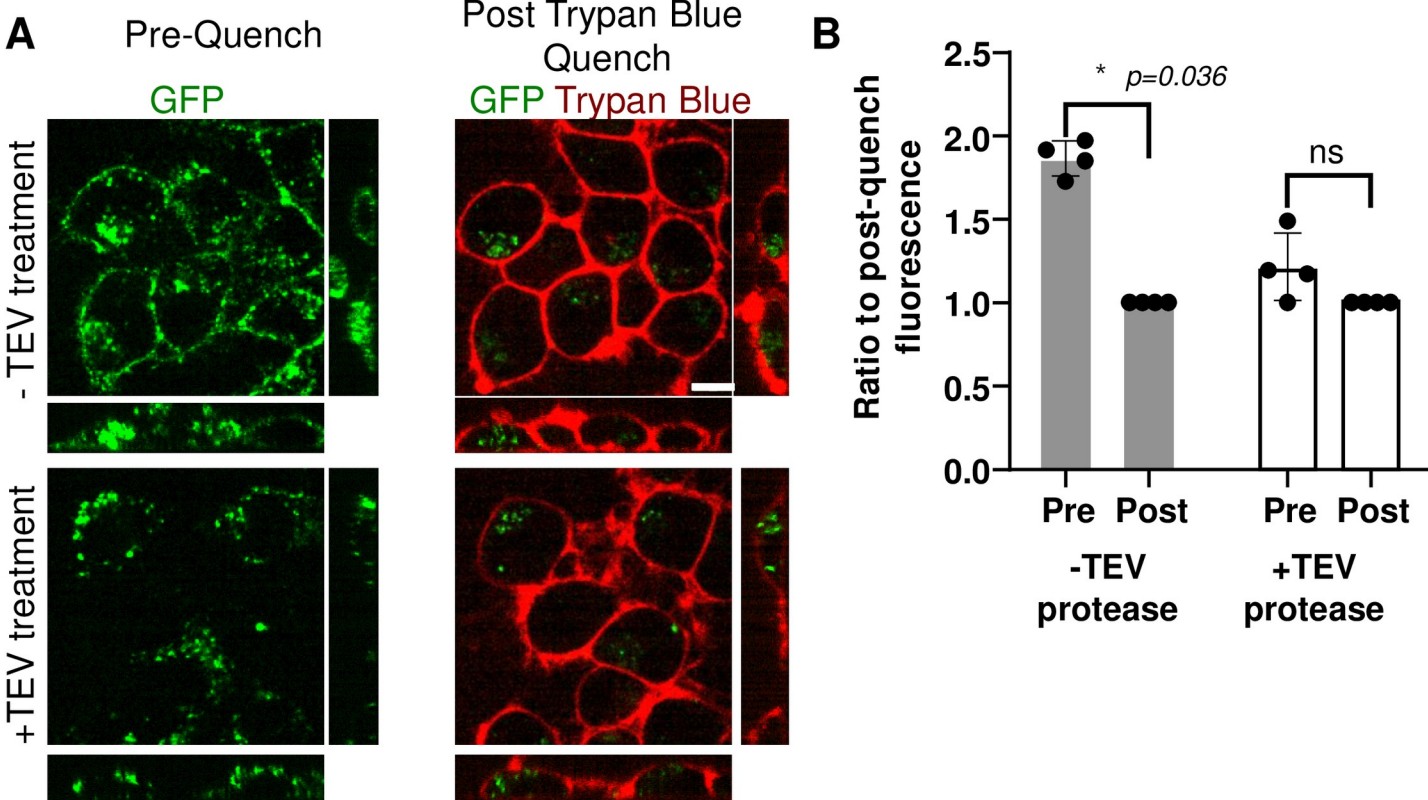

**Fig 3. Incubation with TEV protease selectively removes GFP fluorescence from the cell surface. (Panel A)** Confocal images of Neuro2A cells after incubation with α-syn-TEV-GFP PFFs (*green*) for 1 h, followed by incubation with (*bottom panels*) or without TEV protease (*top panels*) or for 30 min. Images are presented as the average of 3 slices from the middle of a confocal Z-stack (Scale bar = 5 μm). **(Panel B)** Quantification of the percent of fluorescence remaining after trypan blue quenching. The total fluorescence signal in the 3D confocal images was measured both pre- and post-trypan-blue treatment in four independent experiments; the average for each experiment is shown as a single data point. Data are presented as a ratio of final post-trypan blue-quenched fluorescence to pre-trypan blue fluorescence for each individual experiment. *, p = 0.036; *ns*, not significant (p>0.05); Sidak's multiple comparison test.

polymer bottom dishes at 15 μg/ml, and cells incubated for 1 h at 37°C to permit endocytosis. Cells were then either treated with TEV-P or with buffer for 30 min. Following cleavage by TEV-P, free GFP is capable of freely diffusing away from the plasma membrane, greatly reducing the membrane-bound GFP signal. **Fig 3A** depicts confocal images of Neuro2A cells incubated with buffer (*top panel*) or TEV-P (*bottom panels*), either before (*left side*) or after (*right side*) trypan blue quenching. To control for heterogeneity in GFP signal across differing treatments, the total amount of GFP fluorescence signal prior to trypan blue addition was normalized to the post-quench GFP signal in each image. Surface fluorescence in the TEV-P-treated cells is clearly reduced as compared to images obtained prior to trypan blue quenching, with little additional loss of fluorescence after quenching. Overall, the level of GFP signal in control-treated cells prior to quenching is approximately 1.9-fold higher than post-trypan blue quenching, whereas in TEV-P treated wells, the signal prior to the quench is not significantly higher than the post-quench signal (**Fig 3B**). We conclude that TEV-P treatment removes extracellular GFP signal as effectively as trypan blue quenching.

While it is theoretically possible that TEV is internalized and acts on α-syn-TEV-GFP inside the cell rather than externally, the short time frame in which cells are exposed to the protease solution renders this unlikely.

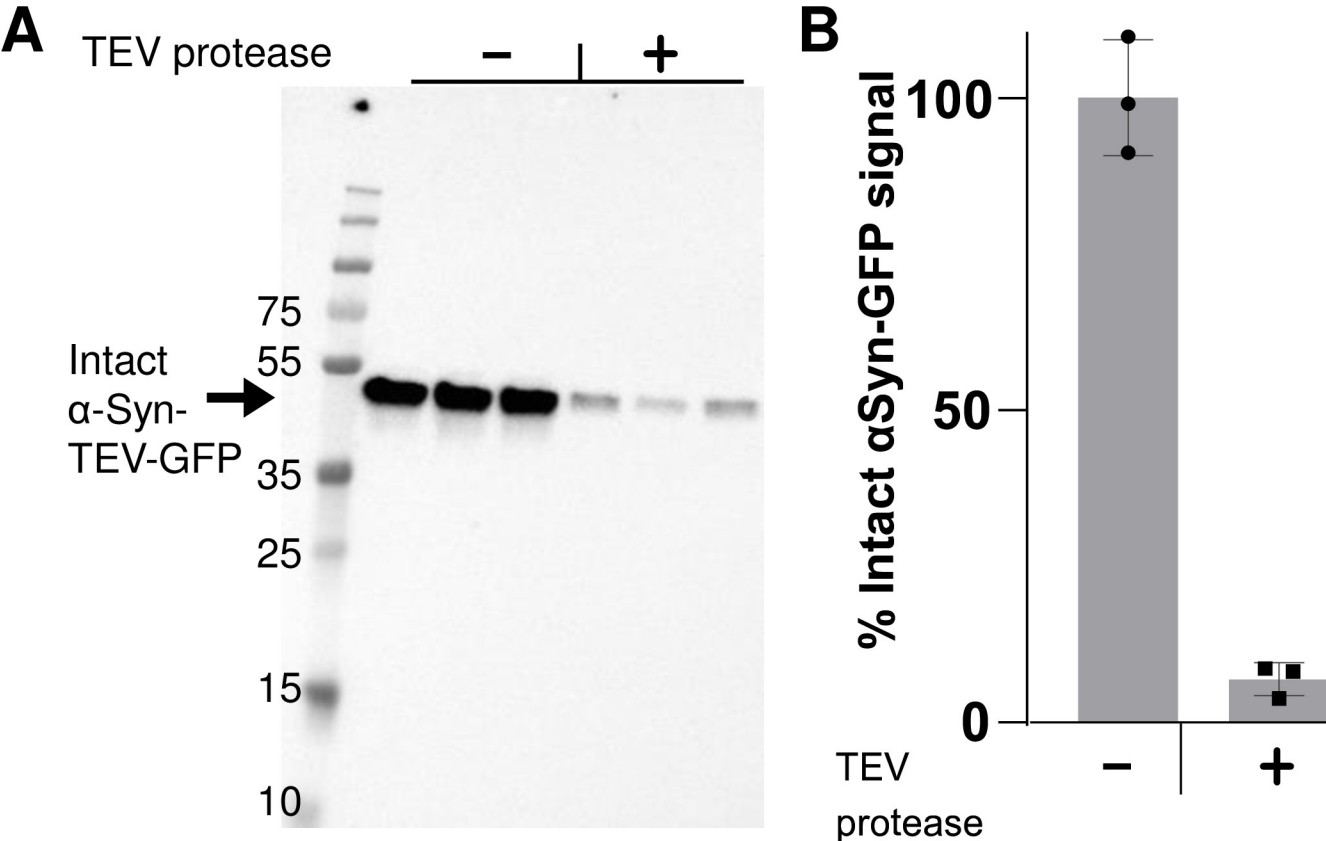

**Fig 4. Western blotting provides a quantitative index of internalized intact PFFs. (Panel A)** Anti-α-synuclein Western blot, showing the total level of intact α-syn-TEV-GFP per well when α-syn-TEV-GFP PFF-treated cells are incubated with or without TEV protease. **(Panel B)** Quantification of Western blot (triplicate samples, mean ± SD).

**Fig 4** presents quantitative Western blotting data which support the utility of the technique in measuring the amount of internalized PFFs. Neuro2A cells were exposed to PFFs for 60 min to allow endocytosis; this was then followed by incubation of cells with TEV-P or buffer control for 60 min. In order to detect total α–syn-TEV-GFP protein levels, preheated 2x sample buffer was added to the wells without washing and wells were heated at 95˚C for 10 min to efficiently monomerize remaining α-syn-TEV-GFP PFFs for SDS-PAGE. These results, showing large quantities of the 45 kDa fusion protein, support the idea that the majority of the GFP seen in fluorescent images derives from surface-associated α-syn-TEV-GFP PFFs (as in **Fig 3**). Exposure of cells to TEV-P removes 95% of this signal (**Fig 4B**), corroborating the efficacy of extracellular TEV treatment and providing a quantitative measure of PFF internalization.

While intracellular expression of TEV-P does not appear to be toxic (reviewed in [12]), we were not able to find any published data on extracellular use of this enzyme. We therefore assessed the potential cytotoxicity of TEV-P treatment using a WST-1 cytotoxicity assay. We found that cellular viability after 30 min of TEV-P treatment and 24 h of recovery was comparable to cells treated only with buffer and to untreated cells (**Fig 5**). These data clearly indicate that exposure to TEV-P is not harmful to cell viability.

**Fig 6A** presents an example of the use of the TEV-P technique to examine co-localization of internalized GFP with the lysosomal dye Lysotracker. In the absence of TEV-P treatment, the intense fluorescence contributed by surface-associated PFFs makes it difficult to detect the signal arising from internalized PFFs, and thereby, their colocalization with Lysotracker.

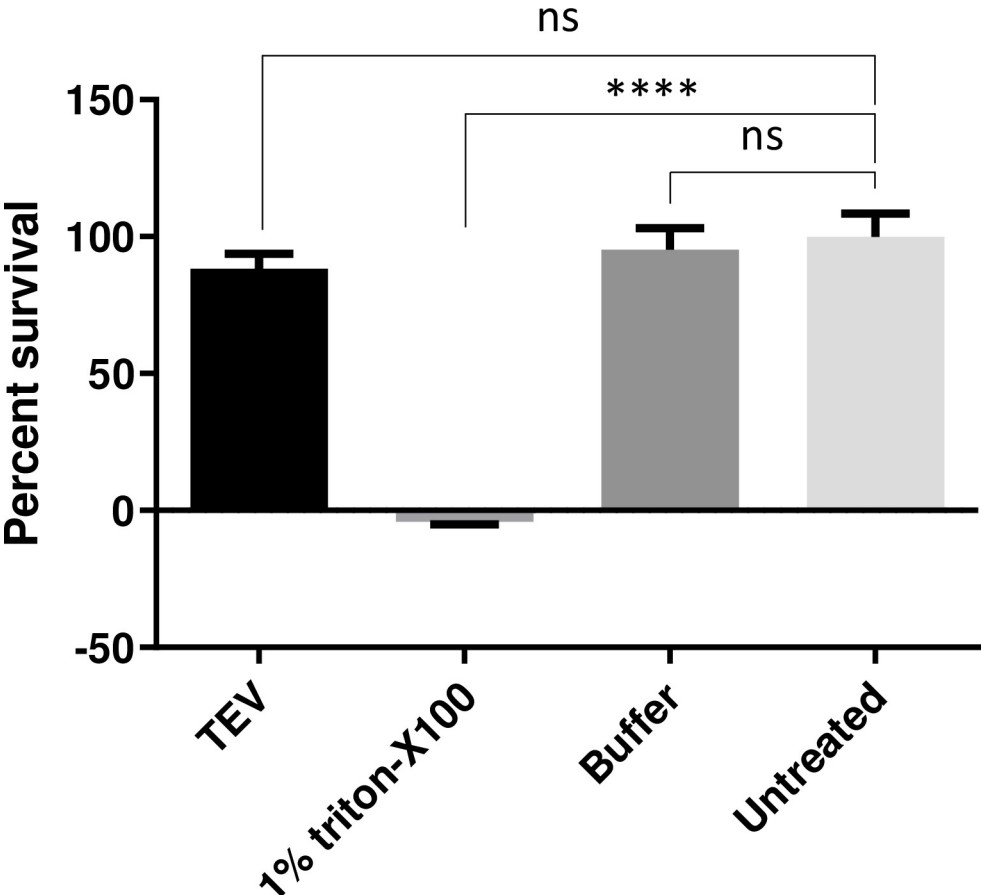

**Fig 5. Exposure of cells to extracellular TEV protease is not cytotoxic.** Neuro2A cells were exposed to TEV-P for 30 min and cell viability measured following an overnight growth period using the WST-1 cytotoxicity assay. Percent survival is plotted for treatment with TEV-P; 1% Triton X-100; 1x TEV-P buffer; and for untreated cells, by normalizing TEV-treated cell absorbance values to untreated cell absorbance values. TEV-P treatment does not result in significant cell death when compared to either buffer-treated or untreated cells. (****, p<0.0001; *ns*, not significant; one-way ANOVA and Tukey's multiple comparisons tests).

Following TEV-P treatment, sparse colocalization of Lysotracker (*red*) with α-syn-TEV-GFP PFFs (*green*) is seen (**Fig 6A**). To control for the effect of TEV-P treatment on the GFP signal, we exposed Neuro2A cells to α-syn-GFP PFFs lacking the TEV-P cleavage site, followed by TEV-P or buffer, and observed no difference in total GFP intensity between the two treatments (**Fig 6B**). However, we found that TEV treatment greatly reduced the total intensity of GFP fluorescence in cells exposed to α-syn-TEV-GFP PFFs as compared to α-syn-GFP PFFs, highlighting the effective use of the TEV-P cleavage site in visualizing internalized GFP PFFs (**Fig 6B**). Quantitation of the total fluorescent intensity of α-syn-TEV-GFP PFFs exposed to TEV-P *vs* buffer treatment showed a significant reduction with TEV-P treatment (**Fig 6B**), confirming the results shown in **Fig 4**.

In **Fig 6C**, imaging of differentiated primary hippocampal neurons exposed to α-syn-TEV-GFP PFFs reveals a small number of internalized PFFs, as shown by co-localization of GFP fluorescence with GFP immunoreactivity (assessed 4 days after PFF treatment). However, primary cells exposed to α-syn-TEV-GFP PFFs and then incubated for 4 days do not show substantive internal GFP fluorescence nor immunoreactivity, nor differences between TEV and buffer treatment. We conclude that our method is not suitable for long-term experiments;

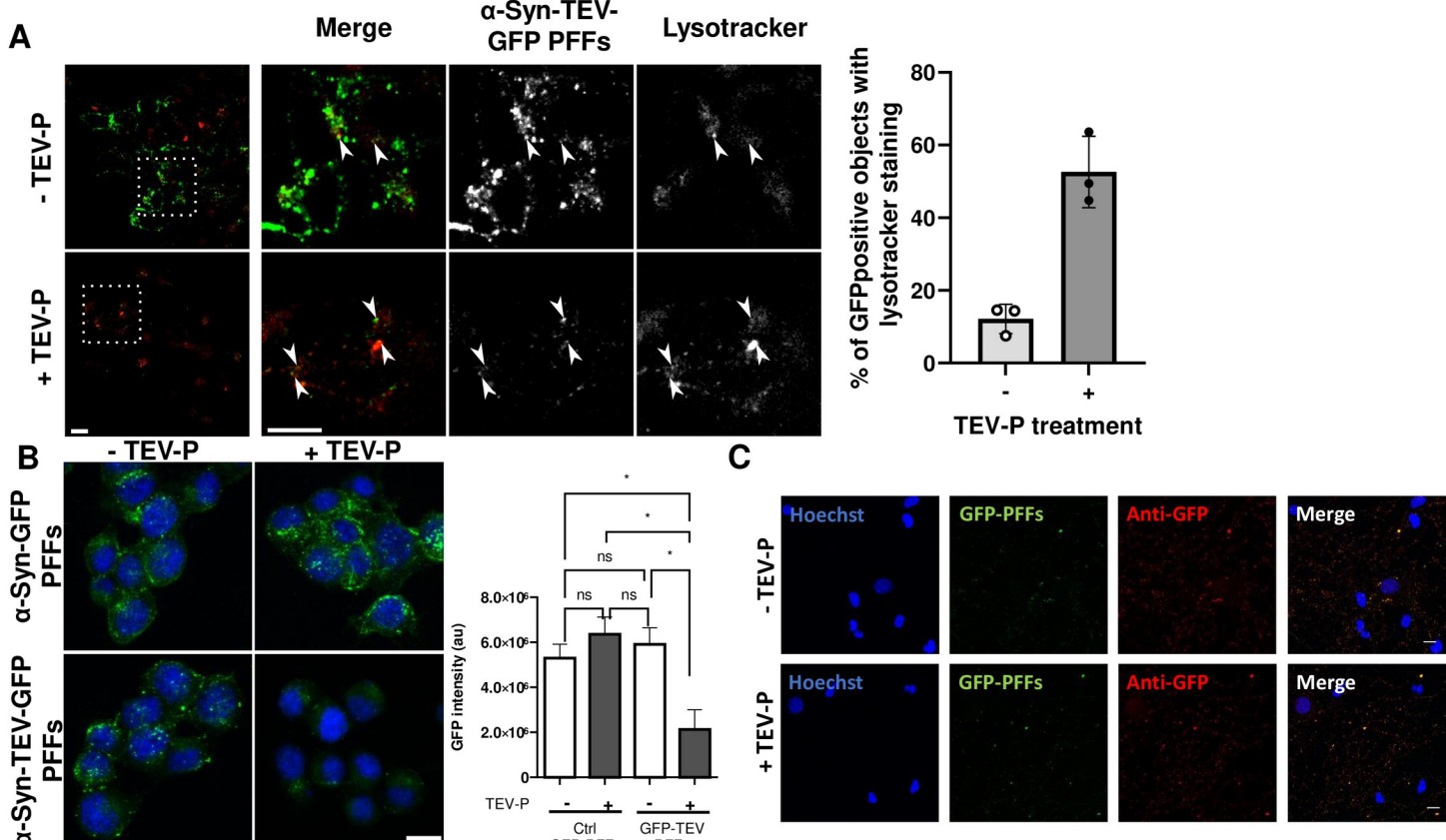

**Fig 6. TEV-P-treated cells can be used for fluorescent imaging of internal markers as well as staining. (Panel A)** Neuro2A cells with Lysotracker. Neuro2A cells incubated with a-syn-TEV-GFP PFFS in the absence of TEV treatment exhibit large quantities of GFP-positive objects that obscure the detection of Lysotracker-positive objects. TEV P-treated Neuro2A cells exhibit a much lower GFP signal, permitting greatly enhanced detection of internalized α-syn-TEV-GFP PFFs colocalized with Lysotracker; *white arrows* indicate colocalization. Scale bar = 5 μm. *Right portion*: Quantitation of GFP-positive internalized objects indicates that co-localization of Lysotracker and GFP is greatly enhanced by TEV treatment. **(Panel B)** Comparison of Neuro2A cells incubated with PFFs either lacking or containing a TEV site. Confocal images were obtained after incubation of cells with either α-syn-TEV-GFP PFFs or α-syn-GFP PFFs (*green*) for 1 h, followed by incubation with TEV protease or buffer for 30 min. Images are presented as the average of 3 slices from the middle of a confocal Z-stack (scale bar = 5 μm). Quantitation of the total fluorescence signal in 3D confocal images was measured for both α-syn-TEV-GFP PFFs and α-syn-GFP PFFs, with and without TEV-P treatment, and shows a clear dependence on the presence of a TEV site (*, p<0.05; ns, not significant; one-way ANOVA and Tukey's multiple comparisons tests). **(Panel C)** Differentiated primary rat hippocampal cells were fixed and stained with anti-GFP antiserum (*red*) 4 days post-exposure to 5 μg/ml α-syn-TEV-GFP PFFs (*green*) and treatment with either buffer (*top row*) or 100 U/ml TEV-P (*bottom row*). Images were taken at 40x; scale bar, 10 μm.

but that it can readily be coupled with post-treatment cell experimentation such as immunocytochemistry.

## Discussion

Mounting evidence suggests that toxic forms of α-synuclein are propagated throughout the peripheral and central nervous systems (reviewed in [1]); thus, the uptake and cellular disposition of α-synuclein is a topic of increasing research interest. At present, it is difficult to establish the cellular location of added PFFs, even when tagged with GFP, because PFFs adsorb so efficiently to cell surfaces. While trypan blue has been successfully used to quench fluorescence arising from surface-associated GFP-PFFs [10], the trypan blue method does not permit further analysis of PFF-treated cells, for example, to establish the cellular itinerary of internalized fluorescent PFFs. The data presented here show that Lysotracker can be used to follow internalized fluorescent GFP-tagged α–synuclein. Additionally, our data show that

immunohistochemical staining of intracellular GFP (as a proxy for synuclein PFFs), and likely of other intracellular markers, can be performed on TEV-treated cells; this is not possible with the trypan blue method. Lastly, while our Western blotting experiment included only a single time point, the disappearance of the internalized intact fusion product over time can potentially be used as a quantitative measure of intracellular PFF turnover.

A key limitation of our method is the fact that TEV treatment removes the fluorescent label but leaves α-synuclein still attached to the cell surface; thus, α-synuclein may still continuously enter the cell while the internalized fluorescent species is under examination, and this internalized α-synuclein may affect cellular processes in parallel with the internalized GFP-labeled PFFs. This is also a limitation of the trypan blue technique: while outside fluorescence is quenched, GFP-PFFs still remain on the cell surface, and are presumably continuously internalized.

The TEV protease method to specifically examine an internalized fluorescent GFP tag can potentially be expanded to include a variety of applications. One example is the use of internal split TEV systems [19] to remove fluorescent signal from internalized PFFs under defined cellular conditions. In addition, while we used a GFP label in this technique as a proof-of-concept, a broad spectrum of protein tags are equally amenable to similar TEV protease protection assays. Fluorophores that are pH-sensitive, photoactivatable, or other biological sensors could theoretically be used in place of the GFP tag (although the efficacy of each tagged synuclein in seeding fibrillation assays would initially need to be tested). In another possible application, affinity tags could be added to permit selective co-immunoprecipitation of intracellular PFFs to screen for intracellular interacting partners (reviewed in [12]). The biocompatibility and specificity of the TEV protease lends itself to a wide variety of potential extensions of this technique. Interestingly, our use of extracellular TEV appears to be a novel application for the use of this protease.

In sum, new tools are required to adequately address the challenges in understanding the cellular events that underlie intercellular α-synuclein spreading. α-Synuclein pathology is based on templated mis-folding of native monomers; a stable seed drastically increases the rate of α-synuclein fibrillation/pathology, and thus the ability to accurately identify the subcellular locations of PFFs during and following the cellular uptake process is critical. Our technique, which permits further biochemical analysis and continued examination of both fixed and live cells after exposure to fluorescent PFFs, represents a strong complement to trypan blue quenching.

## Supporting information

**S1 Raw images.**
(TIF)

## Acknowledgments

Sample preparation and electron microscopy was performed by Dr. Ru-Ching Hsia in the UMB Electron Microscopy Core, while live cell imaging was performed in the UMB Imaging Core. We thank Ms. Minerva Contreras for the preparation of primary hippocampal cells, and Dr. Virginia Lee (University of Pennsylvania) for the pRK172-mouse α-synuclein-GFP expression plasmid.

## Author Contributions

**Conceptualization:** Timothy S. Jarvela.

**Data curation:** Iris Lindberg.

**Formal analysis:** Timothy S. Jarvela.

**Funding acquisition:** Iris Lindberg.

**Investigation:** Timothy S. Jarvela, Kriti Chaplot.

**Methodology:** Timothy S. Jarvela.

**Project administration:** Iris Lindberg.

**Supervision:** Iris Lindberg.

**Writing – original draft:** Timothy S. Jarvela.

**Writing – review & editing:** Iris Lindberg.

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
