## [Decision Letter · Decision Letter 0]

10 Nov 2020

PONE-D-20-31683

A protease protection assay for the detection of internalized alpha-synuclein pre-formed fibrils

PLOS ONE

Dear Dr. Lindberg,

Thank you for submitting your manuscript to PLOS ONE. After careful consideration, we feel that it has merit but does not fully meet PLOS ONE’s publication criteria as it currently stands. Therefore, we invite you to submit a revised version of the manuscript that addresses the points raised during the review process.

1) Please address reviewer #1's concern that "the authors should add a short discussion of the fact that this approach does not exclude a possibility for the α-synuclein PFFs with the removed GFPs to interact with the cell, be internalized, and affect cellular processes in parallel with the internalized GFP-labeled PFFs."

2) Reviewer #2 answered "partly" to the questions "Is the manuscript technically sound, and do the data support the conclusions?" This review asked for several controls. Please address major points 1 and 3-5, as well as the minor points. Addressing these issues will improve your study.

We look forward to receiving your revised manuscript.

Kind regards,

Stephan N. Witt, Ph.D.

Academic Editor

PLOS ONE

Journal Requirements:

i) Please provide an amended statement that declares *all* the funding or sources of support (whether external or internal to your organization) received during this study, as detailed online in our guide for authors at http://journals.plos.org/plosone/s/submit-now.  Please also include the statement “There was no additional external funding received for this study.” in your updated Funding Statement.

ii) Please include your amended Funding Statement within your cover letter. We will change the online submission form on your behalf.

Reviewers' comments:

Reviewer's Responses to Questions

**Comments to the Author**

1. Is the manuscript technically sound, and do the data support the conclusions?

Reviewer #1: Yes

Reviewer #2: Partly

2. Has the statistical analysis been performed appropriately and rigorously? 

Reviewer #1: Yes

Reviewer #2: Yes

3. Have the authors made all data underlying the findings in their manuscript fully available?

Reviewer #1: Yes

Reviewer #2: Yes

4. Is the manuscript presented in an intelligible fashion and written in standard English?

Reviewer #1: Yes

Reviewer #2: Yes

5. Review Comments to the Author

Reviewer #1: This is an interesting and important study describing a useful methodology for generation of the α-synuclein pre-formed fibrils (PFFs) suitable for the analysis of the cellular processes underlying cell-to-cell transmission of α-synuclein proteopathic aggregates. This is achieved by the incorporation of a tobacco etch virus (TEV) protease cleavage site between α-synuclein and green fluorescent protein. Utilization of this construct allows for the efficient removal of GFP from the non-internalized PFFs. Although the presented data are convincing, the authors should add a short discussion of the fact that this approach does not exclude a possibility for the α-synuclein PFFs with the removed GFPs to interact with the cell, be internalized, and affect cellular processes in parallel with the internalized GFP-labeled PFFs.

Reviewer #2: In the present manuscript, Jarvela et al provided a new tool to detect the endocytosed pre-formed alpha-synuclein fibrils. Given the importance of cell-to-cell transmission of alpha-synuclein Pff in Parkinson’s disease pathophysiology, it is interesting to develop the method visualizing transmitted Pff at the cellular level for further mechanism research. However, it is still many critical flaws to reach the author’s conclusion, in particular, missing many control experiment. Also, considering the relatively short text length, there are many mislabeling everywhere.

Major points:

1. The authors used alpha-synuclein construct inserted with His, TEV-p site, and GFP, and then tested pre- and post-TEV treatment effect in most experiments. Since TEV-p site is critical for this new tool development, it should be included No-TEV-p site construct as a negative control.

2. In Fig3, the authors conclude that TEV-P treatment removes extracellular GFP signal as effectively as trypan blue quenching. However, despite the statistical insignificance in Fig 3-B, it looks like a huge difference between pre- and post-quenching after TEV treatment (Fig 3-A bottom image). It would be necessary to show that the GFP positive signals in the pre-quench (+TEV) is not a membrane-localized signal using a membrane marker antibody.

3. In Fig 4, the authors claim that exposure of cells to TEV-P removes 95% of alpha-syn-TEV-GFP signals using western blot. But it may possible that TEV-P protein can enter the cells and induce proteolysis of the endocytosed proteins. Therefore, it needs to prove that TEV-P treatment is solely working on the extracellular absorbed proteins using a negative control expression. As the authors discussed, if this method can be useful for the endocytosed Pff assay tool, the TEV-Protease should not enter cells freely.

4. In Fig6-A, there is no quantification data.

5. In Fig6-B, considering long exposure of pff on the cultured neurons, authors should include the TEV-P no treatment set as a negative control with quantification data.

Minor points:

-In many places, authors used a-syn or a-synuclein instead of α-syn or α-synuclein

-In many places, the authors used both “ⅹ” and “X” (e.g. 40X oil). Need to make it consistently.

-In Fig 5, there is no statistical marks.

-line 66, need to check “ENLYFQˇG”

-line 74, 50ug/ml change to 50μg/ml

-line 110, 37° C change to 37°C

-line 110, CO2 change to CO2 (lower case)

-line 135, 4 C change to 4°C

-line 141, 35-mM change to 35-mm

-line 159, 164, 260, 37 C change to 37°C

-line 276, incorrect labeling of (top) and (bottom)

6. PLOS authors have the option to publish the peer review history of their article (what does this mean?). If published, this will include your full peer review and any attached files.

Reviewer #1: **Yes: **Vladimir N. Uversky

Reviewer #2: No

---

## [Author Response · Author response to Decision Letter 0]

29 Dec 2020

See responses below:

Dear Dr. Lindberg,

Thank you for submitting your manuscript to PLOS ONE. After careful consideration, we feel that it has merit but does not fully meet PLOS ONE’s publication criteria as it currently stands. Therefore, we invite you to submit a revised version of the manuscript that addresses the points raised during the review process.

1) Please address reviewer #1's concern that "the authors should add a short discussion of the fact that this approach does not exclude a possibility for the α-synuclein PFFs with the removed GFPs to interact with the cell, be internalized, and affect cellular processes in parallel with the internalized GFP-labeled PFFs."

We have added this caveat to the manuscript.

2) Reviewer #2 answered "partly" to the questions "Is the manuscript technically sound, and do the data support the conclusions?" This review asked for several controls. Please address major points 1 and 3-5, as well as the minor points. Addressing these issues will improve your study.

We have addressed points 1, 4 and 5 (see rebuttal). We have no method to rebut point 4 as we do not have "a negative control expression". While the TEV enzyme is His-tagged, which would theoretically allow us to examine intracellular TEV uptake, the substrate protein is also His-tagged; thus there is no means to directly examine TEV uptake. As stated in our rebuttal, we feel that intracellular proteolysis is quite remote given the fact that cells are exposed to dilute protease for only 30 min. Intracellular TEV proteolysis would imply a rapid and robust uptake of enzyme followed by substantive internal encounter with PFF substrate in the crowded intracellular environment. This reaction is unlikely to be quantitatively important compared to the readily available surface-associated cleavage reaction.

 A rebuttal letter that responds to each point raised by the academic editor and reviewer(s). You should upload this letter as a separate file labeled 'Response to Reviewers'. DONE

 A marked-up copy of your manuscript that highlights changes made to the original version. You should upload this as a separate file labeled 'Revised Manuscript with Track Changes'. DONE

 An unmarked version of your revised paper without tracked changes. You should upload this as a separate file labeled 'Manuscript'. DONE

---

## [Decision Letter · Decision Letter 1]

13 Jan 2021

A protease protection assay for the detection of internalized alpha-synuclein pre-formed fibrils

PONE-D-20-31683R1

Dear Dr. Lindberg,

We’re pleased to inform you that your manuscript has been judged scientifically suitable for publication and will be formally accepted for publication once it meets all outstanding technical requirements.

Kind regards,

Stephan N. Witt, Ph.D.

Academic Editor

PLOS ONE

Additional Editor Comments (optional):

Reviewers' comments:

Reviewer's Responses to Questions

**Comments to the Author**

1. If the authors have adequately addressed your comments raised in a previous round of review and you feel that this manuscript is now acceptable for publication, you may indicate that here to bypass the “Comments to the Author” section, enter your conflict of interest statement in the “Confidential to Editor” section, and submit your "Accept" recommendation.

Reviewer #2: All comments have been addressed

2. Is the manuscript technically sound, and do the data support the conclusions?

Reviewer #2: Yes

3. Has the statistical analysis been performed appropriately and rigorously? 

Reviewer #2: Yes

4. Have the authors made all data underlying the findings in their manuscript fully available?

Reviewer #2: Yes

5. Is the manuscript presented in an intelligible fashion and written in standard English?

Reviewer #2: Yes

6. Review Comments to the Author

Reviewer #2: (No Response)

7. PLOS authors have the option to publish the peer review history of their article (what does this mean?). If published, this will include your full peer review and any attached files.

Reviewer #2: No

---

## [Editor Report · Acceptance letter]

18 Jan 2021

PONE-D-20-31683R1 

A protease protection assay for the detection of internalized alpha-synuclein pre-formed fibrils 

Dear Dr. Lindberg:

I'm pleased to inform you that your manuscript has been deemed suitable for publication in PLOS ONE. Congratulations! Your manuscript is now with our production department. 

Kind regards, 

on behalf of

Dr. Stephan N. Witt 

Academic Editor

PLOS ONE